# GPattern-Bench: Benchmarking Gene Spatial Pattern Classification in Subcellular Spatial Transcriptomics

## Abstract

Subcellular transcriptomics technologies have revolutionized our ability to study gene expression and its spatial context at single-cell resolution. One fundamental yet underexplored task is *gene spatial pattern classification*, which involves predicting localization patterns for genes within a single cell. To this end, we introduce *GPattern-Bench*, a novel benchmark for this task that unifies evaluation across four established baselines on three diverse datasets, comprising 43 million RNA molecules across 101,000 cells. Given the suboptimal performance of existing machine learning methods, we also develop *GPSNet*, a transformer-based architecture tailored for efficient modeling of spatial transcriptomics data. To address the computational challenges of modeling thousands of RNA molecules in a single cell, we propose a KNN-attention mechanism as a plug-in module for the transformer architecture, enabling the model to efficiently capture spatial dependencies. Extensive experiments on GPattern-Bench demonstrate that GPSNet outperforms existing methods by a significant margin in both accuracy and inference speed, achieving an average F1-macro score of 70% across the three datasets, a relative improvement of over 30% compared to the best baseline. We believe GPattern-Bench will facilitate future research in this area, and GPSNet can serve as a strong deep-learning baseline for future methods. We will publicly release GPattern-Bench and GPSNet to the community.

## 1 Introduction

Spatial transcriptomics (ST) has revolutionized our understanding of cellular organization by enabling the measurement of gene expression while preserving spatial localization information (Williams et al., 2022; Moses & Pachter, 2022; Tian et al., 2023). Recent technological advances, such as MERFISH (Chen et al., 2015), CosMx (NanoString Technologies, 2024), and PHOTON (Rajachandran et al., 2025), have further improved imaging resolution to the subcellular level, as fine as $0.1 \sim 0.2$ $\mu$m. This unprecedented resolution provides researchers with a better understanding of the intricate spatial distribution patterns of individual gene mRNAs within cells, where distinct spatial patterns reflect functional specialization and cellular organization (Cassella & Ephrussi, 2022; Benjamin et al., 2024).

Gene spatial pattern classification is an emerging but underexplored problem that quantitatively classifies the spatial distribution of mRNA molecules for each gene within individual cells. As illustrated in Figure 1, given a set of mRNA molecules with their spatial coordinates and gene identities in a single cell, the goal of this multi-label classification problem is to classify each gene into one or more predefined spatial patterns, such as *nuclear*, *cytoplasmic*, *membrane*, or *granular*. Accurately identifying these patterns is essential for understanding cellular functions, including localized protein synthesis and polarity establishment (Lawrence & Singer, 1986); developmental processes, such as asymmetric cell division (Taliaferro et al., 2014; Martin & Ephrussi, 2009); and disease mechanisms, where disrupted mRNA localization contributes to neurodegeneration (Romo et al., 2018) and other pathological conditions (Taliaferro et al., 2014).

To this end, we propose **GPattern-Bench**, a comprehensive benchmark for gene spatial pattern classification in subcellular spatial transcriptomics. To our knowledge, this is the first benchmark

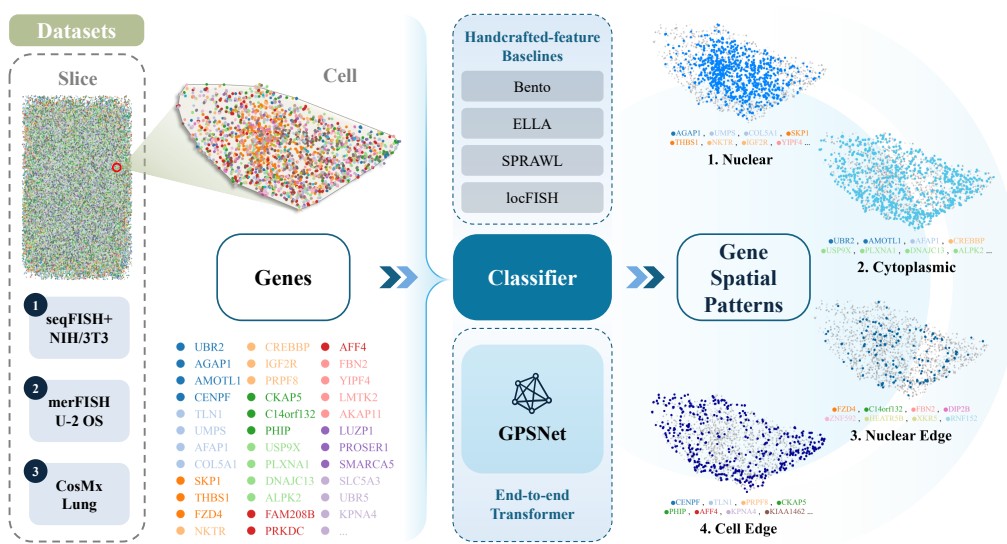

Figure 1: **Gene spatial pattern classification in subcellular spatial transcriptomics.** The task is to perform multi-label classification on genes within a single cell, assigning them to predefined spatial patterns such as *nuclear*, *cytoplasmic*, or *granular* based on their mRNA locations. To address this, we introduce **GPattern-Bench**, a comprehensive benchmark built from three high-resolution datasets and four baselines, and propose **GPSNet**, a novel transformer-based model that sets a new state-of-the-art in performance and efficiency.

specifically designed for this task. The benchmark is constructed from three high-resolution datasets from diverse biological contexts: NIH/3T3 fibroblast cells (Eng et al., 2019), U-2 OS cells (Mah et al., 2024), and CosMx Lung tissue (He et al., 2022). These datasets vary significantly in scale, gene counts, spatial patterns, and species, providing a robust testbed for evaluating computational methods. In total, the benchmark includes **43 million** RNA molecules in **101,000** cells. To facilitate direct comparisons, we adapt and evaluate four representative baseline methods on this benchmark: (a) **Bento** (Mah et al., 2024), which leverages random forests and tensor decomposition to define subcellular domains; (b) **ELLA** (Wang & Zhou, 2024), a probabilistic framework that uses non-homogeneous Poisson processes along unified radial coordinates to estimate spatial patterns via kernel-based intensity functions; (c) **SPRAWL** (Bierman et al., 2024), a non-parametric statistical method that computes rank and permutation-based scores for gene spatial patterns; and (d) **loc-FISH** (Samacoits et al., 2018), which extracts hand-crafted spatial features and employs random forest classification to identify RNA localization patterns. We also implement a unified evaluation protocol using metrics such as AUC, Accuracy, F1-score, and inference speed to facilitate direct comparisons.

Furthermore, we identified that existing machine learning methods show suboptimal performance in both prediction accuracy and inference speed on this task. For example, Bento (Mah et al., 2024) relies on predefined pattern categories and requires accurate nuclear boundary annotations, which can introduce bias and depend on segmentation quality. ELLA (Wang & Zhou, 2024) uses probabilistic modeling but struggles with scalability for extremely large datasets due to its computational intensity with kernel optimization. SPRAWL (Bierman et al., 2024) is limited to specific rank and permutation-based statistical tests, restricting its flexibility for detecting novel or complex spatial patterns. locFISH (Samacoits et al., 2018) depends on hand-crafted features that may not capture intricate spatial relationships beyond predefined classes. Fundamentally, these methods either restrict analysis to human-crafted features or fail to leverage the full spatial context available in modern high-resolution datasets by using pairwise distance metrics or grid-based representations.

Self-attention mechanisms, which have shown powerful capabilities for capturing long-range dependencies in language and vision tasks (Vaswani et al., 2017; Dosovitskiy et al., 2020), offer a promising direction for modeling complex spatial dependencies among entire cells containing hun-

dreds of thousands of molecules in an end-to-end manner. Hence, we propose **GPSNet**, an end-to-end method relying on an encoder-decoder transformer architecture specifically designed to overcome these limitations. The key innovation is our KNN-attention mechanism, which reduces the quadratic complexity of standard self-attention by restricting each RNA molecule to attend only to its K-nearest neighbors. Our comprehensive experiments demonstrate that GPSNet consistently sets a new state-of-the-art, achieving the highest prediction accuracy in terms of AUC, Accuracy, and F1-score, along with fast inference speed across all three datasets, when compared with previous machine learning-based methods.

Our contributions include:

1. **GPattern-Bench**: We formalize the problem of gene spatial pattern classification and propose the first benchmark for this task. The benchmark is curated from three high-resolution subcellular datasets with diverse biological contexts and scales.

2. **GPSNet**: We propose a novel transformer architecture with KNN-attention that efficiently captures local and global spatial dependencies among RNA molecules, enabling accurate classification of gene spatial patterns. The model demonstrates superior classification performance and inference efficiency compared to existing methods.

3. **Experiments and Baselines**: We adapt and evaluate four representative baseline methods on this benchmark with a comprehensive experimental setup, providing a unified evaluation protocol for this field.

## 2  GPATTERN-BENCH

### 2.1  PROBLEM SETTINGS

We formalize gene spatial pattern classification at the level of detected genes within each cell. Let $V$ denote the gene vocabulary and $C$ the set of spatial pattern classes. A cell $i$ contains a variable number $L_i$ of RNA molecules, represented as a set $R_i = \{r_{i,1}, r_{i,2}, \ldots, r_{i,L_i}\}$, where each molecule $r_{i,\ell}$ is a tuple $(g_{i,\ell}, \mathbf{p}_{i,\ell})$ with:

- $g_{i,\ell} \in V$: the gene identity/class of the molecule;

- $\mathbf{p}_{i,\ell} = (x_{i,\ell}, y_{i,\ell}) \in \mathbb{R}^2$: the 2D coordinates of the molecule in a cell-centric frame.

Define the gene-presence mask for cell $i$ as $\mathbf{m}_i \in \{0,1\}^{|V|}$, where $m_i[j] = 1$ if and only if gene $v_j$ is detected in $R_i$ (i.e., $\exists \ell : g_{i,\ell} = v_j$). The learning target for cell $i$ is the gene-pattern pairs binary mask $\mathbf{Y}_i \in \{0,1\}^{|V| \times |C|}$ where:

$$Y_i[j,k] = \begin{cases} 1 & \text{if gene } v_j \text{ exhibits pattern } c_k \text{ in cell } i, \\ 0 & \text{otherwise.} \end{cases} \tag{1}$$

Multiple patterns may be true for the same gene, making it a multi-label classification task. Given a set of RNA molecules $R_i$ in a cell $i$, the model predicts scores $\hat{\mathbf{Y}}_i \in [0,1]^{|V| \times |C|}$ for all gene–pattern pairs in that cell. In practice, binarized predictions are obtained by thresholding the scores per class.

### 2.2  METRICS

We evaluate multi-label performance with several metrics, which are first computed for each pattern class independently and then macro-averaged. This approach ensures a balanced evaluation despite potential class imbalance.

**Notation.** Let there be $N$ instances (present cell–gene pairs) and $C$ pattern classes. For each instance $i$ and class $c$, the ground-truth label is $y_{ic} \in \{0,1\}$ and the model's predicted score is $s_{ic} \in [0,1]$. For each class $c$, let $TP_c, FP_c, TN_c, FN_c$ be the true/false positive/negative counts aggregated over all $N$ instances.

Table 1: **Summary of benchmark datasets.** Avg. RNAs/cell indicates the average number of RNA molecules per cell.

| Dataset | # RNA | # Cell | # Gene | Avg. RNAs/Cell | # Pattern | Technology | Specie |
|---------|-------|--------|--------|----------------|-----------|------------|--------|
| NIH/3T3 (Eng et al., 2019) | 2,724,808 | 179 | 3,721 | 15,222 | 4 | seqFISH+ | Mouse |
| U-2 OS (Mah et al., 2024) | 10,634,467 | 1,022 | 130 | 10,405 | 4 | MERFISH | Human |
| CosMx Lung (He et al., 2022) | 30,370,769 | 100,149 | 960 | 303 | 3 | CosMx SMI | Human |
| **Total** | **43,730,044** | **101,350** | - | **431** | - | - | - |

**Area Under the ROC Curve (AUC).** For each class $c$, we compute the Receiver Operating Characteristic (ROC) curve by varying the decision threshold over the scores $s_{ic}$. The Area Under this Curve, $\text{AUC}_c$, is reported. It is probabilistically equivalent to the likelihood that a randomly chosen positive instance receives a higher score than a randomly chosen negative one. If a class contains only positive or only negative instances in the evaluation set, its AUC is undefined.

**F1-score and Accuracy.** For threshold-dependent metrics, we first determine an optimal, class-specific threshold $\tau_c^*$ that maximizes the F1-score. This is achieved by evaluating the precision-recall curve for each class.

1. For each class $c$, we compute the precision-recall curve from the scores $s_{ic}$ and labels $y_{ic}$.
2. We calculate the F1-score at each point on the curve and identify the threshold $\tau_c^*$ that yields the maximum F1-score, $\text{F1}_c^*$.
3. Using this optimal threshold $\tau_c^*$, we binarize the predictions ($\hat{y}_{ic} = \mathbb{1}[s_{ic} \geq \tau_c^*]$) and compute the corresponding Accuracy ($\text{ACC}_c$).

The final reported metrics are $\text{F1}_{\text{macro}}$, $\text{ACC}_{\text{macro}}$, and $\text{AUC}_{\text{macro}}$, which are the means of the respective per-class scores.

### 2.3 DATASET CONSTRUCTION AND OVERVIEW

We curated three high-resolution spatial transcriptomics datasets to establish a robust benchmark for gene spatial pattern classification. These datasets vary in scale, species, and spatial resolution, providing a comprehensive evaluation ground for our proposed method.

**NIH/3T3 fibroblast cells (Eng et al., 2019).** This dataset profiles $3,721$ genes in $179$ murine fibroblast cells using seqFISH+ technology, with a total of 2.7M molecules. The dataset captures 4 distinct spatial patterns labeled by Bento (Mah et al., 2024): *cell edge, cytoplasmic, nuclear, nuclear edge*. This dataset features a very high number of RNAs per cell ($\sim$ 15k), presenting a challenge for efficient inference and long-context transformer training.

**U-2 OS cells (Mah et al., 2024).** Containing 10M molecules across 1,022 human osteosarcoma cells measured by MERFISH (Chen et al., 2015), this dataset includes 130 genes and 4 spatial patterns labeled by Bento (Mah et al., 2024): *cell edge, cytoplasmic, nuclear, nuclear edge*. It represents a balanced benchmark with a relatively high number of RNAs per cell.

**CosMx Lung (He et al., 2022).** This large-scale human lung tissue dataset contains over 30M molecules across 100k cells from a cancerous tissue, profiling 980 genes with 3 spatial patterns: *nuclear, membrane, cytoplasmic*, using CosMx SMI (NanoString Technologies, 2024). Its massive scale tests the scalability and accuracy of computational methods.

Numerical statistics of each dataset are summarized in Table 1. And, more details about data preprocessing and pattern annotation are provided in Appendix B.2.

### 2.4 BASELINES

We adapted four state-of-the-art methods as baselines for comparison in our GPattern-Bench: (a) **Bento** (Mah et al., 2024), (b) **ELLA** (Wang & Zhou, 2024), (c) **SPRAWL** (Bierman et al.,

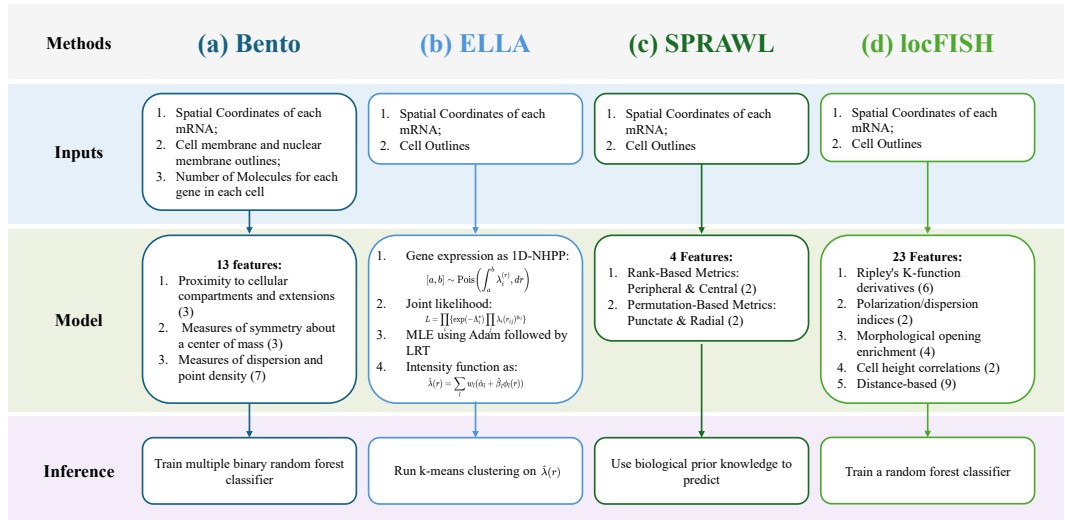

Figure 2: **Methodology of baselines.** a) Bento extracts 13 spatial features from ST data with both cell and nuclear boundaries, and leverages random forests for pattern classification, b) ELLA models gene expression as a 1D NHPP with Adam-optimized intensity functions and performs clustering to derive results, c) SPRAWL computes 4 rank/permutation metrics to score localization patterns, and d) locFISH derives 23 spatial features and applies random forest classification.

2024), and (d) **locFISH** (Samacoits et al., 2018). In Figure 2, we illustrate and compare these four baselines in terms of the data they use as inputs , how they model gene expression, and how they perform inference from the modeling.

We notice that ELLA, SPRAWL, and locFISH use only RNA coordinates and cell outlines as input, while Bento requires additional information such as RNA and cell/nuclear boundaries. Bento, SPRAWL, and locFISH extract handcrafted spatial features from the input data and apply random forest classification or directly predict using biological prior knowledge, whereas ELLA models gene expression as a 1D non-homogeneous Poisson point process (NHPP) with Adam-optimized intensity functions and performs clustering to derive results. Detailed (re)implementations for each baseline are provided in Appendix B.2.1.

## 3 GPSNET

To address the low prediction accuracy and slow inference speed of existing methods, we propose the **G**ene **P**attern **S**patial **Net**work (GPSNet), shown in Figure 3. GPSNet is a novel transformer-based architecture specifically designed to model the complex interactions between gene identity and the spatial context of RNA molecules within single cells. Its key innovation is the introduction of a KNN-attention mechanism that enables efficient modeling of local spatial relationships while maintaining scalability to large numbers of molecules. It is an end-to-end deep learning model that does not rely on handcrafted features and effectively leverages modern GPU hardware.

### 3.1 ARCHITECTURE

**Components.** GPSNet utilizes a modern encoder-decoder transformer-based architecture designed to effectively capture the large number of complex interactions between gene identity and the spatial arrangement of RNA molecules within single cells. GPSNet employs five core components: (1) a coordinate encoding MLP, (2) a gene embedding layer, (3) a transformer encoder with KNN-attention, (4) a transformer decoder with interleaved self-attention and cross-attention, and (5)

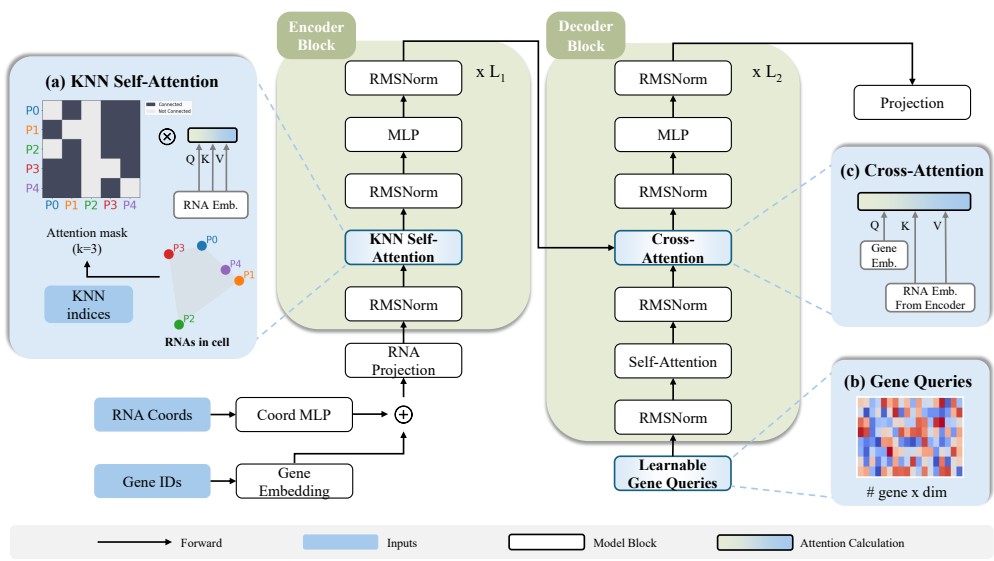

Figure 3: **The architecture of GPSNet.** The model takes RNA molecules, defined by their gene type and spatial coordinates, as input. These are converted into embeddings and processed by a transformer encoder-decoder structure. The final output is a multi-label prediction of gene spatial patterns. The key components of our architecture are: (a) **KNN Self-Attention** in the encoder to efficiently model spatial relationships between neighboring molecules, (b) **Learnable Pattern Queries** in the decoder that act as prototypes for different spatial patterns, and (c) **Cross-Attention**, which allows the decoder to aggregate gene embeddings from the encoder's RNA representation of the cell. Residual connections in the transformer blocks are omitted for clarity.

a multi-label classification head. RMSNorm (Zhang & Sennrich, 2019) and GELU (Hendrycks & Gimpel, 2016) activations are used throughout the model.

**Forward Pass.** The forward pass through GPSNet consists of four stages:

1. **Embedding**: The coordinates and gene classes of each RNA molecule are projected into a shared embedding space using the coordinate MLP and gene embedding layer, respectively. These are summed to create initial representations: $\mathbf{H}^{(0)} = \{\mathbf{h}_1^{(0)}, \mathbf{h}_2^{(0)}, \ldots, \mathbf{h}_L^{(0)}\}$.

2. **Encoder**: The transformer encoder processes these representations using $N$ layers of KNN-attention to capture spatial contexts. Each layer updates the representations as $\mathbf{H}^{(l)} = \text{EncoderLayer}(\mathbf{H}^{(l-1)})$.

3. **Decoder**: The transformer decoder takes learnable gene queries $\mathbf{Q} = \{\mathbf{q}_1, \mathbf{q}_2, \ldots, \mathbf{q}_C\}$ (where $C$ is the number of gene classes) and attends to the encoder outputs through cross-attention: $\mathbf{O} = \text{Decoder}(\mathbf{Q}, \mathbf{H}^{(N)})$. Self-attention within the decoder allows gene queries to interact. These two types of attention are interleaved in each decoder layer.

4. **Classification**: The decoder outputs are passed through a classification head consisting of a linear layer followed by a sigmoid activation to produce multi-label predictions: $\hat{\mathbf{Y}} = \sigma(\mathbf{W}\mathbf{O} + \mathbf{b})$.

### 3.2 KNN-ATTENTION MECHANISM

KNN-attention is proposed to handle two unique challenges of spatial transcriptomics data:

**2D/3D Spatial Structure of Molecules.** RNAs, unlike natural language tokens or image pixels, are points that exist in continuous 2D or 3D space. Biologically, different parts of cells can have very distinct biological functions, and RNAs located far apart are less likely to influence each other.

Therefore, global self-attention, which allows every molecule to attend to every other molecule, is not biologically appropriate. KNN-attention addresses this by restricting attention to local neighborhoods, allowing the model to focus on spatially relevant interactions.

**Large Number of Molecules in a Single Cell.** A single cell can contain thousands to tens of thousands of RNA molecules, making the $O(L^2)$ complexity of standard self-attention infeasible. This is the main computational challenge in applying vanilla transformers to large sets of RNA molecules within a cell. The KNN-attention mechanism addresses the $O(L^2)$ complexity bottleneck of standard self-attention by reducing it to $O(LK)$, making it feasible to process complex and large cells with over 10,000 molecules.

## 4 EXPERIMENTS

### 4.1 MAIN RESULTS

Table 2: **Performance comparison across datasets and methods.** Best results are **bolded** and second best are underlined. For a fair comparison, we report the validation time of our method and ELLA using one GPU with a batch size of 1. *NIH/3T3 and U-2 OS were labeled by Bento; hence, Bento is not evaluated on these datasets. †For ELLA evaluation on the CosMx Lung, due to computational constraints with the large dataset scale (∼100,000 cells), we conducted random sampling for approximation.

| Dataset | Method | AUC (%) ↑ | ACC (%) ↑ | F1-Score (%) ↑ | Validation Time ↓ |
|---|---|---|---|---|---|
| NIH/3T3 | Bento* | - | - | - | 7 min 38 s |
| | ELLA | 50.16 | 77.35 | 11.11 | 23 hr 53 min 24 s |
| | SPRAWL | 67.33 | 80.43 | 23.58 | 5 min 9 s |
| | locFISH | 62.53 | 65.55 | 32.90 | 13 min 20 s |
| | GPSNet (Ours) | **90.22** | **84.35** | **71.02** | **15 s** |
| U-2 OS | Bento* | - | - | - | 7 min 8 s |
| | ELLA | 50.17 | 78.36 | 6.29 | 2 d 16 hr 26 min |
| | SPRAWL | 68.94 | 78.98 | 21.09 | 27 min 21 s |
| | locFISH | 60.17 | 63.25 | 34.15 | 1 d 8 hr 13 min |
| | GPSNet (Ours) | **92.52** | **87.88** | **73.48** | **26 s** |
| CosMx Lung | Bento | 65.92 | 68.18 | 51.72 | 2 d 3 hr 40 min |
| | ELLA† | 65.57 | 53.36 | 25.27 | > 100 d |
| | SPRAWL | 54.01 | 60.27 | 42.86 | 1 hr 2 min |
| | locFISH | 73.42 | 69.76 | 49.69 | 2 d 13 hr 51 min |
| | GPSNet (Ours) | **83.09** | **78.08** | **68.26** | **3 min 6 s** |

Table 2 presents a comprehensive performance comparison between GPSNet and the four baseline methods across all three datasets. GPSNet consistently outperforms all baselines on every metric and dataset, demonstrating its effectiveness for gene spatial pattern classification.

In terms of accuracy, no single baseline consistently performs best. For instance, locFISH achieves the highest F1-score among baselines on NIH/3T3 and U-2 OS, while Bento is the top performer on CosMx Lung, suggesting that existing methods may be specialized to certain data characteristics. In contrast, GPSNet demonstrates superior and robust performance across the board. It achieves a 30.6% relative F1-score improvement over Bento on the challenging CosMx Lung dataset, a 125.9% improvement over locFISH on NIH/3T3, and a 113.1% improvement over locFISH on U-2 OS.

Regarding inference speed, GPSNet is significantly faster than all baselines across all datasets. Baseline methods exhibit a wide range of inference times, from several minutes to multiple days, with none approaching GPSNet's efficiency. On the smallest dataset, NIH/3T3, GPSNet completes inference in just 15 seconds, while the next fastest, SPRAWL, takes over 5 minutes. On the largest dataset, CosMx Lung, where SPRAWL cannot perform in a reasonable time, GPSNet finishes the evaluation in 3 minutes. This efficiency, combined with its high accuracy, makes GPSNet a highly scalable and practical solution for analyzing large and complex spatial transcriptomics data. These

results suggest that GPSNet's integrated approach provides a more comprehensive solution for gene spatial pattern classification.

## 4.2 ABLATION STUDY

We conducted extensive ablation studies on the NIH/3T3 and U-2 OS datasets to understand the contributions of GPSNet's key components. Detailed numerical results of the ablation study are provided in Appendix D.

**Value of K in KNN-attention.** We investigated the impact of the neighborhood size $K$ in KNN-attention. We observed that performance generally improves as $K$ increases. For U-2 OS, performance peaks at $K = 256$ and then declines, suggesting that an excessively large neighborhood can introduce noise. For the NIH/3T3 dataset, performance ceases to improve at $K = 64$ and slightly drops at $K = 256$. However, to maintain consistency across datasets and balance performance with computational efficiency, we selected $K = 256$ as the value for all experiments, as it offers a strong trade-off.

**Encoder-Decoder Depth Ratio.** We studied the effect of the encoder-decoder depth ratio. To ensure a fair comparison, we kept the total number of layers $N$ constant while varying the ratio of encoder to decoder layers. We found that for both NIH/3T3 and U-2 OS, an unbalanced '2:1' or '1:2' ratio shows similar performance and is better than the balanced '1:1' configuration. This suggests that a slightly asymmetrical architecture can be effective. Our final models use the 2:1 ratio for all datasets.

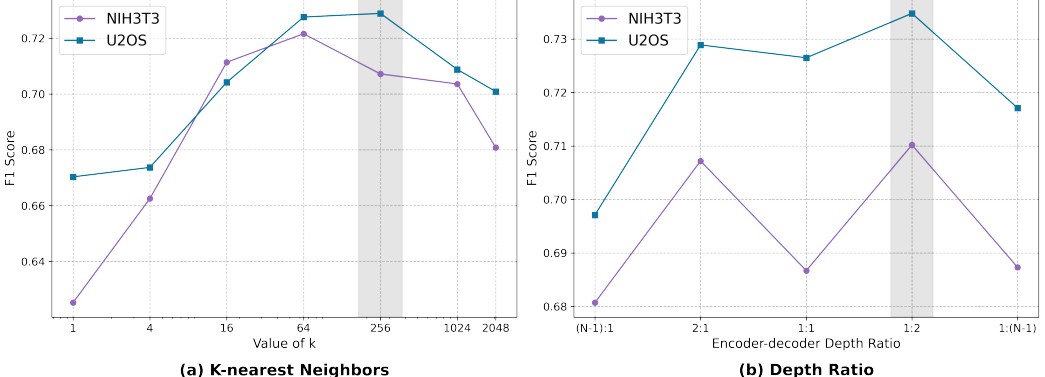

Figure 4: **Ablation studies of GPSNet.** (a) **Effect of KNN neighborhood size K.** We chose $K = 256$ for all datasets to balance performance and efficiency. (b) **Effect of encoder-decoder depth ratio.** We chose a 1:2 encoder-decoder layer ratio for optimal performance. Our choice is in the gray box.

## 4.3 VISUALIZATION

In Figure 5, we show a representative visualization result comparing GPSNet predictions with ground truth and baseline methods on a sample from the U-2 OS dataset.

## 5 RELATED WORK

**Gene Spatial Pattern Classification.** Recent technological advancements in spatial transcriptomics have dramatically improved spatial resolution from tissue-level to subcellular-level measurements, with technologies like MERFISH, SeqFISH+, and VisiumHD achieving resolutions as fine as 0.1-2.0 $\mu m$. This enables the precise measurement of gene expression at the subcellular level, creating new computational challenges for analyzing intracellular mRNA localization patterns. Early methods focused on tissue-level analysis, such as identifying spatially variable genes (Review &

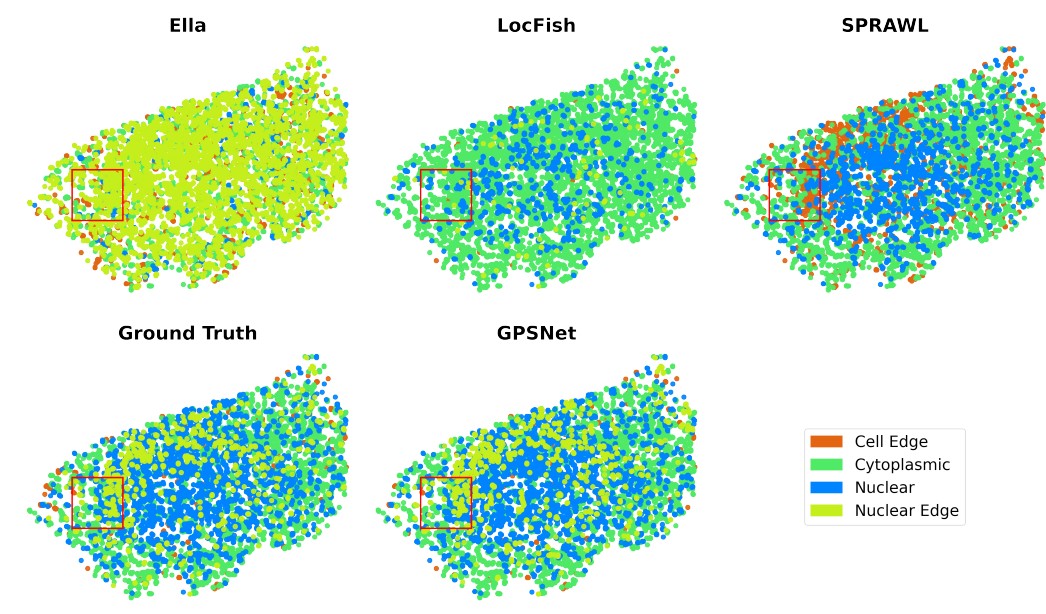

Figure 5: **Comparison of gene spatial pattern prediction on a U-2 OS sample on ground truth and four methods.** GPSNet more accurately recovers the complex pattern geometry compared to baseline methods.

Methods, 2024), but are insufficient for subcellular analysis. Current subcellular methods can be categorized into feature-based approaches like Bento (Mah et al., 2024) and locFISH (Samacoits et al., 2018), which extract hand-crafted features for classification, and statistical modeling approaches like SPRAWL (Bierman et al., 2024) and ELLA (Wang & Zhou, 2024), which use mathematical models to quantify spatial patterns. GPattern-Bench provides a systematic evaluation framework that highlights the respective strengths and weaknesses of these state-of-the-art approaches.

**Transformers in Biology.** Transformers have revolutionized computational biology, demonstrating remarkable success in protein structure prediction (Jumper et al., 2021), single-cell analysis (Yang et al., 2022), genome analysis (Avsec et al., 2025), and spatial omics (Yang et al., 2022). Their self-attention mechanism is uniquely powerful for modeling long-range dependencies, making them theoretically ideal for spatial data. However, the $O(L^2)$ complexity of standard transformers has limited their application to large-scale spatial transcriptomics datasets, where $L$ (the number of RNA molecules) can exceed 100,000 per sample. Recent adaptations using graph-based transformers (Madhu et al., 2025) have attempted to address this. Our GPSNet introduces a physically-grounded KNN-attention mechanism inspired by sliding-window attention (Beltagy et al., 2020) that leverages the inherent spatial locality of biological interactions, providing an efficient and interpretable solution for modeling subcellular spatial transcriptomics data.

## 6 CONCLUSION

We introduced GPattern-Bench, a comprehensive benchmark for gene spatial pattern classification, featuring three diverse datasets and four strong baselines to facilitate future comparisons. To improve prediction accuracy and efficiency, we designed GPSNet, a novel transformer architecture with KNN-attention that effectively handles large-scale spatial transcriptomics data. Extensive experiments demonstrate that GPSNet outperforms all baseline methods across all datasets and metrics, with particularly significant gains on large-scale data, in terms of both prediction accuracy and inference speed. We hope our benchmark and model will stimulate future research in spatial transcriptomics analysis, especially at the subcellular resolution.

## ETHICS STATEMENT

This research utilizes publicly available datasets. The cell lines used in our benchmark (NIH3T3, U-2OS and CosMx Lung) are standard, commercially available cell lines. All data is fully anonymized and was obtained from previously published studies. No new data was collected for this study. The advancements in spatial transcriptomics analysis from this research have the potential to accelerate the understanding of complex biological processes and diseases, such as cancer and developmental disorders. By providing more accurate and efficient tools for analyzing gene expression patterns within their spatial context, this work could ultimately contribute to the development of novel diagnostic methods and therapeutic strategies, improving human health.

## REPRODUCIBILITY STATEMENT

To ensure the reproducibility of our results and to facilitate future research, the source code for our proposed GPSNet model, the implementation of the baseline methods, and the complete GPattern-Bench benchmark datasets will be made publicly available. The repository will include the processed datasets, training scripts, evaluation scripts, and detailed instructions for reproducing the experiments presented in this paper. All hyperparameters and model configurations are described in the experimental section of this paper.

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

APPENDIX

This appendix provides the following additional information:

- Appendix A: Use of LLM.

- Appendix B: Implementation details for datasets, baselines, and hardware/software configurations.

- Appendix C: Hyperparameters for training GPSNet.

- Appendix D: Numerical results of the ablation study on GPSNet.

- Appendix E: Biological meanings of the spatial patterns used in the GPattern-Bench dataset.

- Appendix F: Class distribution of gene spatial patterns in each dataset of GPattern-Bench.

- Appendix G: Additional visualizations of gene spatial pattern predictions on all datasets.

## A    USE OF LLM STATEMENT

We used a Large Language Model (LLM) to assist with grammar checking.

## B    IMPLEMENTATION DETAILS

### B.1    HARDWARE AND SOFTWARE CONFIGURATIONS

All experiments were conducted on a server equipped with four NVIDIA A40 GPUs. All models were implemented using PyTorch 2.7.

### B.2    DATASETS IMPLEMENTATION

The NIH/3T3, U-2 OS, and CosMx Lung datasets are all publicly accessible. Cell and nuclear boundaries for the NIH/3T3 and U-2 OS datasets were obtained from previously processed segmentations (Mah et al., 2024). For the CosMx Lung dataset, we performed cell and nuclear segmentation from the corresponding TIFF images and assigned the resulting boundaries to cells within each field of view.

We randomly partitioned the datasets into training, validation, and test sets using an 80:10:10 ratio based on fields of view. This approach ensures that the methods must demonstrate cross-slice generalization capabilities and effectively handle batch effects across different imaging regions. Statistics are shown in Figure 6.

#### B.2.1    BASELINES IMPLEMENTATION

To ensure a fair comparison, all baseline methods were evaluated using standardized preprocessing in which benchmark spatial transcriptomics datasets (seqFISH+, MERFISH, CosMx) were converted to extract molecular coordinates, cell boundaries, and nucleus boundaries when available.

**SPRAWL.** SPRAWL was installed via PyPI and required converting data into HDF5 format. SPRAWL computes three scores per gene-cell pair: peripheral (tendency toward the cell membrane), central (tendency toward the cell center), and punctate/radial (clustering patterns). These continuous scores were mapped to our discrete labels using thresholds: high peripheral and low central scores indicated cell edge localization; low peripheral and high central scores indicated nuclear localization; significantly negative peripheral and non-significant central scores indicated cytoplasmic patterns; and significantly positive radial and punctate scores indicated polarized localization. This mapping is supported by established knowledge of subcellular RNA localization patterns (Buxbaum et al., 2015) and the SPRAWL methodology (Bierman et al., 2024).

**ELLA.** ELLA was implemented using Poetry for dependency management, following the official documentation. Data preparation involved converting AnnData objects to pickle files and then generating JSON files via command-line tools. ELLA outputs five clusters that were mapped to our categories: the red cluster (nuclear), the yellow cluster (nuclear edge), the green cluster (cytoplasmic), the blue cluster (cell membrane), and the fifth cluster (mixed patterns). When implementing ELLA on the CosMx lung dataset, the large scale of test data ($\sim$100,000 cells) presented computational challenges, with an estimated runtime exceeding 100 days for the full dataset. To make the analysis feasible, we performed random sampling to reduce the dataset to 2,000 cells. This sampling approach is reasonable since ELLA processes one gene at a time to generate consistent spatial patterns across all cells.

**Bento.** Bento was installed via bento-tools with careful dependency management. The implementation followed the Data Prep Guide to format transcript coordinates and cell segmentations for Bento's spatial analysis pipeline.

**locFISH.** locFISH was implemented through the big-fish package, which was originally designed for microscopy images. We adapted it for Spatial Transcriptomics using the same two-stage strategy as the original: first applying t-SNE dimensionality reduction followed by K-means clustering for gene-cell pair classification, and then aggregating patterns across cells for gene-level profiling. The feature extraction pipeline computes spatial statistics directly from transcript coordinates rather than from pixel intensities.

All methods were evaluated using identical metrics, including multi-label accuracy, per-category precision/recall, and overall classification performance, with consistent ground truth labels to ensure reproducible comparisons.

## C  HYPERPARAMETERS FOR TRAINING GPSNET

With our hardware configuration, the training times for the NIH/3T3, U-2 OS, and CosMx Lung datasets are approximately 5 minutes, 25 minutes, and 2 hours, respectively. We provide the detailed hyperparameters used for training GPSNet in Table 3.

Table 3: **Hyperparameters for GPSNet across datasets.**

| Config \ Dataset | NIH/3T3 | U-2 OS | CosMx Lung |
|---|---|---|---|
| encoder depth | 4 | 4 | 6 |
| decoder depth | 8 | 8 | 12 |
| attention heads | | 12 | |
| hidden dim | | 768 | |
| mlp ratio | | 4 | |
| model size | 130M | 130M | 187M |
| optimizer | | AdamW | |
| learning rate | | 1e-5 | |
| weight decay | | 0.03 | |
| optimizer momentum | | (0.9, 0.9) | |
| batch size | 4 | 4 | 64 |
| learning rate schedule | | linear warmup then constant | |
| warmup steps | 1000 | 1000 | 5,000 |
| epochs | 10 | 5 | 10 |
| precision | | bfloat16 | |
| max grad norm | | 1.0 | |
| gradient checkpointing | True | False | False |

# D  ABLATION STUDY

This section provides the numerical results of our ablation studies on K in KNN-attention in Table 4 and on the encoder-decoder depth in Table 5.

Table 4: **Ablation study of GPSNet on K in KNN-attention.** We fixed the encoder-decoder depth ratio to 1:2.

| Dataset | K | AUC (%) ↑ | ACC (%)↑ | F1-Score (%)↑ |
|---------|------|-----------|----------|---------------|
| NIH/3T3 | 1 | 84.00 | 77.13 | 62.52 |
|         | 4 | 87.08 | 80.46 | 66.25 |
|         | 16 | 90.05 | 83.88 | 71.14 |
|         | 64 | 90.77 | 84.84 | 72.16 |
|         | 256 | 89.28 | 83.11 | 70.72 |
|         | 1024 | 89.64 | 83.16 | 70.36 |
|         | 2048 | 88.12 | 81.63 | 68.08 |
| U-2 OS | 1 | 88.44 | 83.04 | 67.03 |
|         | 4 | 88.70 | 83.74 | 67.37 |
|         | 16 | 90.77 | 85.64 | 70.42 |
|         | 64 | 92.20 | 86.78 | 72.76 |
|         | 256 | 92.45 | 86.63 | 72.89 |
|         | 1024 | 91.69 | 86.58 | 70.88 |
|         | 2048 | 90.95 | 85.47 | 70.09 |

Table 5: **Ablation study of GPSNet on encoder-decoder depth ratio.** We fix the value of K to 256.

| Dataset | E:D Ratio | AUC (%) ↑ | ACC (%)↑ | F1-Score (%)↑ |
|---------|-----------|-----------|----------|---------------|
| NIH/3T3 | (N-1):1 | 88.02 | 81.63 | 68.07 |
|         | 2:1 | 89.28 | 83.11 | 70.72 |
|         | 1:1 | 88.18 | 81.43 | 68.67 |
|         | 1:2 | 90.22 | 84.35 | 71.02 |
|         | 1:(N-1) | 88.23 | 81.68 | 68.73 |
| U-2 OS | (N-1):1 | 90.29 | 85.66 | 69.71 |
|         | 2:1 | 92.45 | 86.63 | 72.89 |
|         | 1:1 | 91.95 | 87.20 | 72.65 |
|         | 1:2 | 92.52 | 87.88 | 73.48 |
|         | 1:(N-1) | 91.41 | 86.34 | 71.71 |

# E  BIOLOGICAL MEANING OF SPATIAL PATTERNS

This section provides a brief biological explanation of the spatial patterns used in the GPattern-Bench dataset:

- **Nuclear**: Genes that are predominantly localized within the cell's nucleus.
- **Nuclear edge**: Genes that are localized at the periphery of the nucleus, often associated with the nuclear envelope.
- **Cell edge**: Genes found at the outer boundary of the cell, often involved in cell signaling and interaction with the extracellular environment.
- **Cytoplasmic**: Genes distributed throughout the cytoplasm, involved in various cellular processes such as metabolism and protein synthesis.
- **Membrane**: Genes associated with the cell membrane, playing roles in transport, signaling, and cell adhesion.

## F    CLASS DISTRIBUTION OF GPATTERN-BENCH

This section shows the distribution of gene spatial patterns in each dataset of GPattern-Bench in Figure 7.

## G    MORE PREDICTION VISUALIZATION

This section provides additional visualizations of predictions on all datasets in Figure 8, Figure 9, and Figure 10.

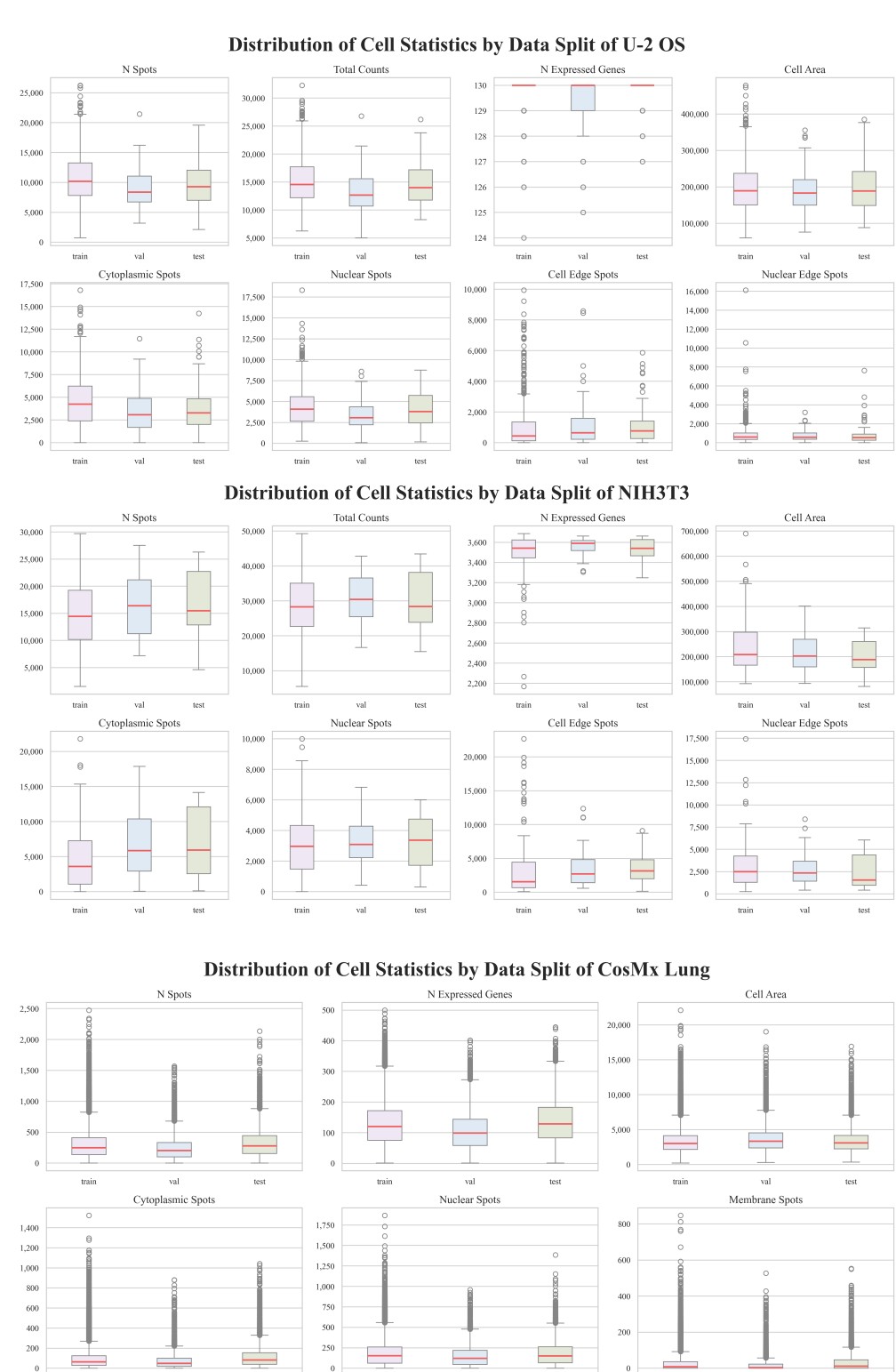

Figure 6: **Statistics of train/validation/test splits for each dataset.** We split each dataset into training, validation, and test sets with a ratio of 80%, 10%, and 10%, respectively. The splits are stratified to ensure that each split contains a representative distribution of gene spatial patterns.

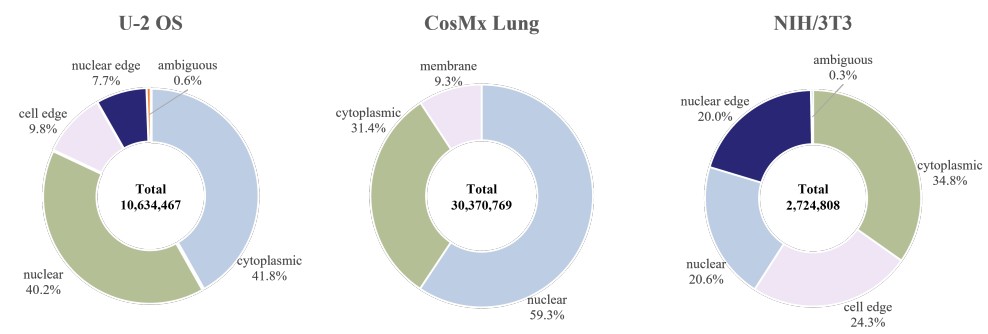

Figure 7: **Distribution of gene spatial patterns in each dataset.** *Ambiguous* genes in U-2 OS and NIH/3T3 are genes that have two spatial patterns.

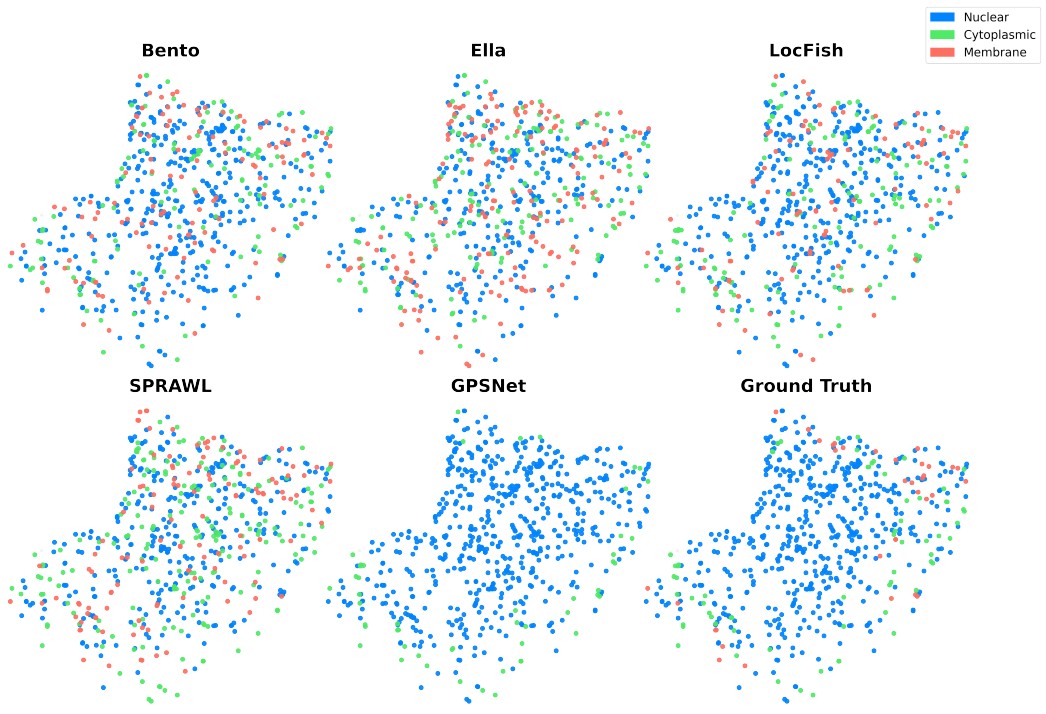

Figure 8: **More comparison of prediction for gene spatial patterns on CosMx Lung.**

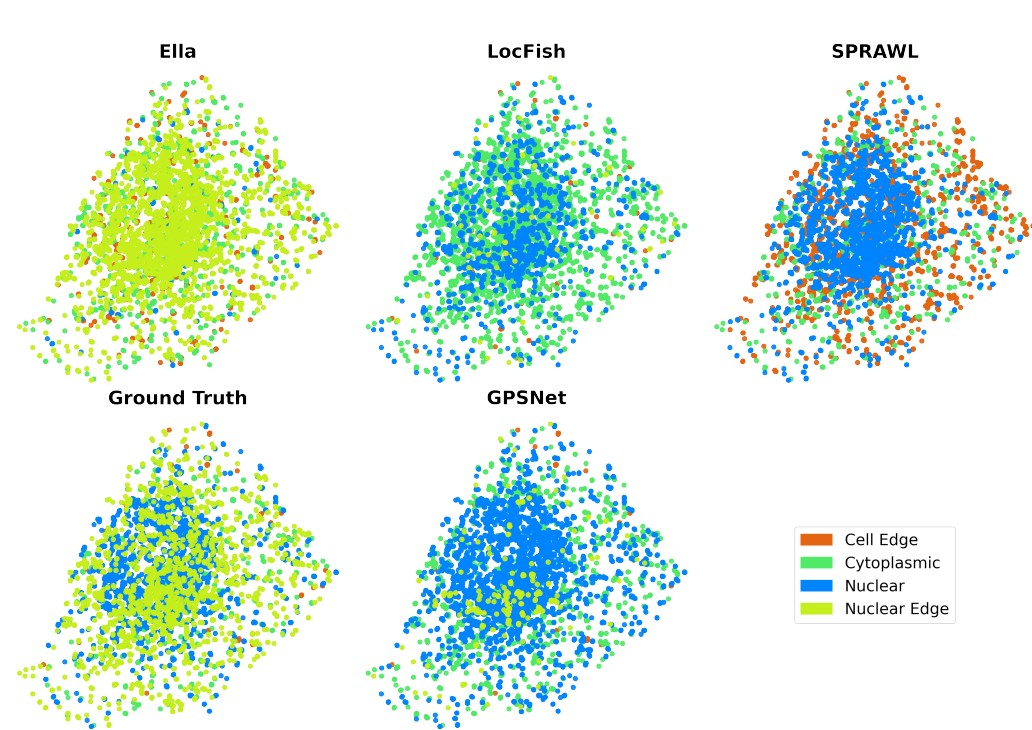

Figure 9: **More comparison of prediction for gene spatial patterns on U-2 OS.**

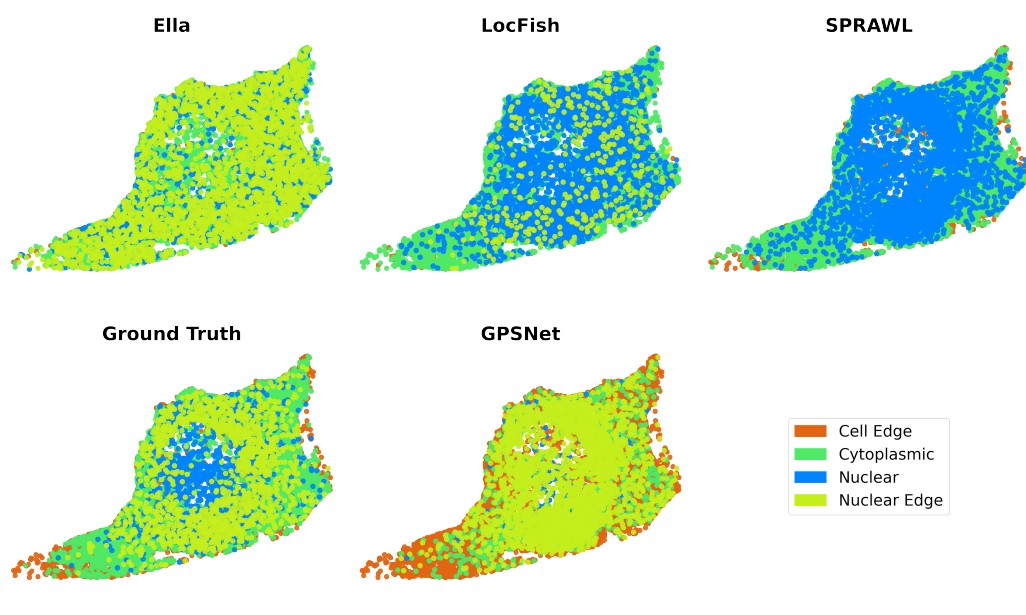

Figure 10: **More comparison of prediction for gene spatial patterns on NIH/3T3.**

