# OpenReview forum: "GPattern-Bench: Benchmarking Gene Spatial Pattern Classification in Subcellular Spatial Transcriptomics"
_ICLR.cc/2026/Conference — ICLR 2026 Conference Withdrawn Submission_

### Official Review · Reviewer_EQHC · 2025-10-15

**Soundness:** 2
**Presentation:** 3
**Contribution:** 2
**Rating:** 2
**Confidence:** 3

**Summary:**

This paper introduces GPattern-Bench, a large-scale benchmark built upon 3 public datasets for subcellular gene localization pattern prediction, which is formulated as a multi-label classification task. The paper also presents GPSNet, a Transformer-based model with a KNN-attention mechanism to efficiently handle spatial transcriptomics data that comprises a large number of genes and their spatial coordinates. Comparison between GPSNet and 4 machine learning baselines shows that the proposed model significantly improves the prediction accuracy and inference speed.

**Strengths:**

- Identifying the subcellular localization of genes with spatial transcriptomics is a novel and underexplored task that plays a significant role in analyzing single cells and building AI virtual cells (AIVC).
- The proposed GPattern-Bench, upon public release, could serve as a valuable resource for fairly evaluating existing approaches.
- The authors also provide an easy but effective approach, i.e., GPSNet, that shows significant improvements over ML-based methods in both accuracy and scalability.
- The paper is generally well-written and easy to follow.

**Weaknesses:**

- The contribution of the proposed benchmark is limited due to its nature as a simple aggregation of existing resources.
  - As acknowledged by the authors, the benchmark is merely a collection of three pre-existing datasets without further post-processing, harmonization, or generation of novel data.
  - Another significant concern is the extreme heterogeneity across these three datasets, which exhibit substantial differences in species, sequencing technologies, spatial resolutions, and the number of single cells and genes. Crucially, the manuscript lacks a deeper, comparative analysis of these heterogeneous datasets, which is necessary to establish their collective utility and to properly frame the challenges they present for a unified modeling approach.
- The proposed GPSNet model exhibits limited technical novelty and questionable architectural soundness.
  - The use of kNN graphs for modeling spatially-resolved data is a well-established practice in the field [1,2], and kNN-based attention mechanisms have been extensively explored in prior studies [3,4]. Consequently, the core mechanisms of the encoder do not represent a significant methodological advance.
  - The design of the decoder appears conceptually unsound. In standard decoder architectures, the attention mechanism employs an upper triangular mask, but the gene queries lack meaningful sequential orders. A more intuitive alternative would be a purely encoder-based architecture combined with an appropriate pooling head operating over the representations of mRNA transcripts belonging to the same gene.
- The presentation of the main results lacks sufficient depth and biological significance.
  - Section 4.1 primarily highlights that the Deep Learning-based GPSNet outperforms traditional Machine Learning approaches, which is foreseeable and rather trivial.
  - In Section 4.2, the benefits of an asymmetrical architecture lack a proper explanation.

Refs.

[1] Decompdiff: diffusion models with decomposed priors for structure-based drug design

[2] MLGCN: An Ultra-Efficient Graph Convolution Neural Model For 3D Point Cloud Analysis

[3] KVT: k-NN Attention for Boosting Vision Transformers

[4] Long-Range Transformers with Unlimited Length Input

**Questions:**

My major concerns have been listed in the weaknesses above. Below are a few additional questions.

Concerning technical details:
- How is the gene embedding obtained? Are they randomly initialized and trained or derived from proprietary embeddings from knowledge graphs [1] or DNA sequences [2]?
- In Line 311, the learnable queries contain $C$ features where $C$ is the number of gene classes. However, in Line 134, $C$ represents spatial pattern classes, leading to ambiguity. Moreover, what's the design rationale of maintaining these queries and applying cross-attention between queries and RNA molecule representations?
- Is MLP a robust choice for encoding RNA coordinates within different cells, which may exhibit distinct shapes? Have the authors investigated alternative SE(2)-invariant architectures [3]?
- In Appendix B.2, the authors mentioned that the dataset is split based on "fields of view". What does it mean by "fields of view", and how does it reflect the challenge of cross-slice generation?
- Do the authors notice data noise like an mRNA is transcribed in the nucleus, but yet to be carried to the cell membrane? If yes, does it affect the robustness of the trained models?

Concerning experiment results:
- Since the distribution of class labels varies across different datasets, I'm wondering if there exists a confusion pattern of ML models that may provide some biological insights.
- Which factor mainly contributes to the model prediction? For example, according to prior biological studies [4], certain genes show clear connections with mRNA localization. Does the model implicitly capture the spatial patterns, like the cell membrane and nuclear membrane, to facilitate prediction?
- How do different choices of K contribute to the scalability of GPSNet? According to the ablation studies, I think that $K=64$ is a more appropriate choice for its competitive performance and potentially improved speed.
- Does the model generalize to novel cells and slices robustly? Have the authors investigated transfer learning settings (e.g., training on one dataset and evaluating on others)?

Refs.

[1] Multimodal reasoning based on knowledge graph embedding for specific diseases

[2] DNA language model GROVER learns sequence context in the human genome

[3] SE(3)-Transformers: 3d roto-translation equivariant attention networks

[4] mRNA localization: Gene expression in the spatial dimension.

---

### Official Review · Reviewer_CdmW · 2025-10-29

**Soundness:** 3
**Presentation:** 2
**Contribution:** 2
**Rating:** 6
**Confidence:** 2

**Summary:**

This paper introduces GPattern-Bench, the first benchmark for gene spatial pattern classification in subcellular spatial transcriptomics data, and proposes GPSNet, a Transformer-based model that employs a KNN-attention mechanism to reduce computational complexity. The paper evaluates four baseline methods on three datasets of varying scales and demonstrates the advantages of GPSNet in both prediction accuracy and inference speed.

**Strengths:**

1.Systematically formalizes the "gene spatial pattern classification" task for the first time and establishes a multi-label classification framework

2.The paper integrates three datasets from different species, technological platforms, and scales, providing strong representativeness and practical utility

3.The paper introduces a KNN-attention mechanism that effectively alleviates the computational bottleneck of standard Transformers when applied to large numbers of RNA molecules

**Weaknesses:**

1.All performance comparisons are presented as point estimates without standard deviations or significance tests, making it impossible to assess the stability of the performance differences.

2.The KNN-attention is essentially a straightforward adaptation of local attention mechanisms, highly similar to those used in GNN or sparse Transformers, lacking substantial theoretical or architectural breakthrough

3.GPSNet is a large-scale model with a substantial number of parameters. However, for datasets such as NIH/3T3 (with only 179 cells) and U-2 OS (with 1,022 cells), there exists a significant risk of overfitting

4.In the NIH/3T3 and U-2 OS datasets, the so-called "ground truth labels" were generated by the baseline method Bento. this introduces a circular dependency problem, the entire benchmark's "ground truth" is itself the output of a machine learning method, rather than being independently biologically validated. It may lead to inherent bias in the evaluation.

**Questions:**

1.Why was KNN chosen over radius-based neighborhood construction? Is there biological evidence supporting that "K-nearest neighbors" is more appropriate than "spatial proximity"?

2.Does the model exhibit systematic bias on extremely imbalanced classes? Were strategies like re-weighting or Focal Loss explored?

---

### Official Review · Reviewer_8K7T · 2025-10-30

**Soundness:** 2
**Presentation:** 2
**Contribution:** 1
**Rating:** 4
**Confidence:** 3

**Summary:**

The paper proposes GPATTERN-BENCH, a benchmark for a novel task that classifies the spatial pattern (nuclear, cytoplasmic, membrane, or granular) of each gene based on a set of mRNA molecules with known spatial coordinates and gene identities within a single cell. Three datasets (NIH/3T3, U-2 OS, and CosMx Lung) are used to curate the benchmark. Moreover, the paper evaluates several baseline methods and introduces a GPSNet based on KNN-attention for the gene spatial pattern classification task.

**Strengths:**

The paper curates a benchmark for the proposed task and evaluates it against multiple baseline methods.

**Weaknesses:**

- Unclear motivation

The classification of nuclear, cytoplasmic, membrane, or granular localization has been widely studied. It is unclear why the authors chose to tackle this task using gene-level spatial data, which is significantly more costly to obtain. The rationale for this approach and its advantages over existing imaging-based methods are not clearly justified.

- The contribution of the paper is unclear to me.

In Section 2.3, the benchmark is described as being assembled from multiple datasets (NIH/3T3, U-2 OS, and CosMx Lung). However, it is unclear whether any specific post-processing, relabeling, or other curation steps were applied to construct the benchmark. If no additional processing was performed, the novelty of the benchmark itself is questionable.

For GPSNet and KNN-attention, the paper emphasizes the KNN-attention mechanism in GPSNet, but it is not clear how this differs from similar approaches in the point cloud literature. The novelty of the proposed method is difficult to discern based on the current description.

- Comparison methods

The methods compared in the paper are mainly non-deep-learning approaches, such as random forest classifiers. This raises the question of whether more advanced baselines, particularly graph neural network (GNN)-based methods, should have been included. While GPSNet demonstrates the best performance on the proposed benchmark, the comparison may not fully establish its superiority over state-of-the-art deep learning or graph-based approaches.

**Questions:**

- Why not use H&E-stained images

While the classification of subcellular patterns (nuclear, cytoplasmic, membrane, granular) has been explored using H&E-stained images in morphological analysis, it is unclear why the authors chose to rely on spatial transcriptomics data, which is more costly, rather than leveraging these established image-based approaches. The value of using gene-level spatial information over image-based methods is not clearly justified.

---

### Official Review · Reviewer_XWqa · 2025-11-01

**Soundness:** 3
**Presentation:** 3
**Contribution:** 3
**Rating:** 4
**Confidence:** 3

**Summary:**

The authors of this paper introduce GPattern-Bench, a benchmark for the task of gene spatial pattern classification in subcellular spatial transcriptomics. The benchmark combines three diverse datasets (NIH/3T3, U-2 OS, CosMx Lung) totaling over 43 million RNA molecules across ~100k cells. They further propose GPSNet, a transformer-based encoder-decoder model with a novel KNN-attention mechanism to address scalability and biological locality. Experiments demonstrate strong improvements: GPSNet achieves ~70% macro F1, a >30% relative improvement over the best baseline, with substantial gains in inference speed.

**Strengths:**

- Unlike general-purpose architectures, GPSNet’s KNN-attention mechanism is biologically grounded, as it can capture local spatial dependencies while maintaining scalability to millions of RNA molecules. This design can address the biological challenges inherent to spatial transcriptomics.

- A strength of the paper is that it clearly defines and grounds a new benchmark for the task of gene spatial pattern classification at the subcellular level. Right now, this problem is fragmented across toolkits (Bento, locFISH, SPRAWL), each with its own assumptions, inputs, and outputs. By introducing GPattern-Bench, the authors formalize the task, specify the data sources, and standardize the evaluation, turning it into something the community can iterate on.

**Weaknesses:**

- While the paper is careful to re-implement classical and statistical baselines (Bento, ELLA, SPRAWL, locFISH), it does not include comparisons against more modern deep learning approaches that could be adapted to this setting, such as point-cloud transformers, set/graph transformers, or recent spatial omics models that already use attention with locality constraints. Since GPSNet is itself a transformer-style model, the absence of such baselines makes it harder to isolate how much of the gain comes from the proposed KNN-attention and task-specific decoder, and how much simply comes from moving from feature engineering to end-to-end learning.

- The work does not yet explore cross-dataset or cross-technology transfer. All results are within-dataset. That setup is easier, because the model can implicitly learn dataset-specific morphology and labeling conventions. A benchmark that claims to be “the first” for this task would be stronger if it also reported train-on-MERFISH, test-on-CosMx; or at least showed how much tuning is needed to move across platforms. Right now, it is not obvious whether GPattern-Bench measures generalization in subcellular ST, or mostly measures performance on three fairly homogeneous splits.

**Questions:**

- You motivate KNN-attention both biologically (locality) and computationally (O(LK)), which is reasonable. Did you try other sparse/structured attention variants with comparable or lower complexity, such as local window attention, dilated/sliding windows or linear-complexity attention mechanisms?

- Several of the baselines you adapt are quite sensitive to cell/nucleus boundary quality. One selling point of a learned model is that it could be more robust. Did you test GPSNet in settings with noisy or incomplete segmentations (e.g. perturbed masks, missing nuclei, slightly misaligned cell outlines)?

- All reported results are intra-dataset. Since your benchmark contains three different technologies (seqFISH+, MERFISH, CosMx) with different spatial densities and label sets, a natural question is: how well does GPSNet trained on one technology transfer to another?

- Right now, the comparison is against four mostly non–deep learning methods (Bento, ELLA, SPRAWL, locFISH). Since GPSNet is a transformer-style model, could you also compare with modern deep neural network baselines?

- In the last few years, we’ve seen large pretrained models for scRNA-seq (such as [1]) and spatial data (cell-level, not molecule-level). Do you think such models could be adapted to your setting, for example by using them to gene-level embeddings for each cell?

[1]  Wen, Hongzhi, et al. "CellPLM: Pre-training of cell language model beyond single cells." BioRxiv (2023): 2023-10.

---

### Note · Authors · 2025-12-03

I have read and agree with the venue's withdrawal policy on behalf of myself and my co-authors.